# Impact of Traditional and Non-Traditional Lipid Parameters on Outcomes after Intravenous Thrombolysis in Acute Ischemic Stroke

**DOI:** 10.3390/jcm11237148

**Published:** 2022-12-01

**Authors:** Chua Ming, Emma M. S. Toh, Qai Ven Yap, Leonard L. L. Yeo, Ching-Hui Sia, Andrew F. W. Ho, Yiong Huak Chan, Fathima Ashna Nastar, Amanda Y. L. Lim, Benjamin Y. Q. Tan

**Affiliations:** 1Yong Loo Lin School of Medicine, National University of Singapore, Singapore 117597, Singapore; 2Department of Biostatistics, Yong Loo Lin School of Medicine, National University of Singapore, Singapore 117597, Singapore; 3Division of Neurology, Department of Medicine, National University Health System, Singapore 119074, Singapore; 4Department of Cardiology, National University Heart Centre, Singapore 119074, Singapore; 5Department of Emergency Medicine, Singapore General Hospital, Singapore 169608, Singapore; 6Pre-Hospital and Emergency Research Centre, Duke-NUS Medical School, Singapore 169857, Singapore; 7Division of Endocrinology, Department of Medicine, National University Health System, Singapore 119074, Singapore

**Keywords:** ischemic stroke, intravenous thrombolysis, tissue plasminogen activator, low-density lipoprotein cholesterol, high-density lipoprotein cholesterol, symptomatic intracranial hemorrhage

## Abstract

Contradicting evidence exists regarding the role of lipids in outcomes following intravenous (IV) thrombolysis with tissue plasminogen activator (tPA). Restricted cubic spline curves and adjusted logistic regression were used to evaluate associations of low-density lipoprotein cholesterol (LDL-C), non-high-density lipoprotein cholesterol (non-HDL-C), total cholesterol (TC), high-density lipoprotein cholesterol (HDL-C) and LDL-C/HDL-C ratio with poor functional outcome, symptomatic intracranial hemorrhage (SICH) and 90-day mortality, among 1004 acute ischemic stroke (AIS) patients who received IV tPA in a comprehensive stroke center. Quartile (Q) 1, Q2 and Q3 of HDL-C were associated with increased odds of poor functional outcome (adjusted odds ratio (adjOR) 1.66, 95% CI 1.06–2.60, *p* = 0.028, adjOR 1.63, 95% CI 1.05–2.53, *p* = 0.027, adjOR 1.56, 95% CI 1.01–2.44, *p* = 0.048) compared to Q4. Q2 and Q4 of non-HDL-C were associated with increased odds of SICH (adjOR 4.28, 95% CI 1.36–18.90, *p* = 0.025, adjOR 5.17, 95% CI 1.64–22.81, *p* = 0.011) compared to Q3. Q1 and Q2 of LDL-C was associated with increased odds of mortality (adjOR 2.57, 95% CI 1.27–5.57, *p* = 0.011 and adjOR 2.28, 95% CI 1.10–5.02, *p* = 0.032) compared to Q3. In AIS patients who received IV tPA, low LDL-C was associated with increased odds of mortality while HDL-C may be protective against poor functional outcome.

## 1. Introduction

Stroke incidence and stroke-related mortality increased substantially from 1990 to 2019, and the stroke burden will continue to increase globally, especially in underdeveloped countries [1]. In Singapore, a study reported an increased prevalence of stroke risk factors and the increased crude incidence rate of stroke among those younger than 65 [2]. There has been conflicting evidence on how lipid parameters affect post-thrombolysis outcomes [3,4], with non-traditional lipid parameters like non-HDL-C suggested to have similar functions as LDL-C in predicting hemorrhagic transformation (HT) [5]. Pre-stroke statin use was also shown to influence ischemic stroke outcomes [6].Hence, we aimed to explore the general associations of selected traditional and non-traditional lipid parameters with post-thrombolysis outcomes, to compare their clinical utility in prognostication. One significance of this study is to facilitate determination of cholesterol target levels in achieving optimal post-stroke recovery. Lastly, there are varied results on the relationship between lipid parameters and large artery atherosclerosis (LAA) stroke subtype. Bang et al. found that higher non-HDL-C and TG, but not LDL-C, was associated with LAA. Several studies have also shown that LDL-C may not best predict atherosclerotic vascular risk [7]. However, Hindy et al. suggested LDL-C lowering likely reduces LAA risk [8]. These should be further assessed to evaluate the utility of targeting lipid parameters for LAA stroke prevention. 

## 2. Materials and Methods

### 2.1. Study Design

In this study, we included consecutive patients who received IV tPA from September 2006 to June 2018. All these patients had no contraindications to IV tPA use. The study obtained ethics approval from the National Healthcare Group-Domain Specific Review Board (NHG DSRB Ref: 2010/00509). Patients were assessed by a neurologist for eligibility to receive intravenous thrombolysis according to institutional protocol and American Heart Association/American Stroke Association guidelines at a standard dose of 0.9 mg/kg body weight [9]. All thrombolysed stroke patients underwent standard non-contrast head computed tomography and computed tomography brain and neck angiography. Patients who were deemed as unsuitable for IV-tPA, or who underwent endovascular thrombectomy were excluded from the study. Patients that were included had valid lipid parameters of TC, HDL-C and TG collected in fasting conditions, in mmol/litre, that were taken within 24 h of their AIS admission as per our institution protocol (Figure 1).

LDL-C was calculated by the Friedewald equation (TC-HDL-C-TG/5) [10], while some had LDL-C collected directly. Non-HDL-C levels were calculated by subtracting HDL-C levels from TC. LDL-C/HDL-C ratio was calculated by dividing LDL-C by HDL-C. Other baseline demographics, clinical parameters and ischemic stroke characteristics were collected and tabulated within 24 h of AIS admission. Diabetes mellitus (DM) was defined as pre-existing diagnosis of diabetes mellitus or an admitting fasting blood glucose level greater than or equal to 7.0 mmol/L or an HbA1c greater than or equal to 6.5% [11]. The severity of stroke at presentation was assessed using the National Institute of Health Stroke Scale (NIHSS) [12]. This assessment was made by credentialed nurses as part of the acute stroke response team. The presence of Large Vessel Occlusion (LVO) was defined as occlusions of the first and second segment of the middle cerebral artery (MCA), M1 and M2, the Internal Carotid Artery (ICA) and as well as its terminus, tandem occlusions involving ICA-MCA, or occlusion of the basilar artery. The Trial of ORG 10172 in Acute Stroke Treatment (TOAST) criteria was used by the treating stroke neurologist to classify stroke subtypes [13]. Investigations to evaluate the TOAST mechanism included: vessel imaging with either computed tomography angiography (CTA), magnetic resonance angiography (MRA), transcranial doppler (TCD), carotid duplex or a combination of these, as well as Holter monitoring and transthoracic echocardiography.

The primary outcome measured was poor functional outcome (90-day modified Rankin Scale (mRS) of 3–6). Secondary outcomes measured were symptomatic intracranial hemorrhage (SICH) and 90-day all-cause mortality. SICH was based on the European Cooperative Acute Stroke Study (ECASS) II definition [14]. The 90-day mRS was evaluated during follow-up visit to the stroke clinic. If not, mRS was evaluated via telephone call instead.

### 2.2. Data Analysis

Analyses were performed using SPSS for Windows version 27.0 (SPSS Inc, Chicago, IL, USA) and R version 4.0.5. Restricted cubic splines with 5 knots were plotted to visually assess the univariate associations between the lipid parameters and the three outcomes. These served qualitative and descriptive purposes only. Since non-linear (U-shaped/inverse U-shaped/J-shaped/reverse tick) associations were found, lipid parameters were divided into quartiles for analysis. Descriptive statistics for continuous variables were presented as mean (SD) when normality and homogeneity assumptions were satisfied, otherwise as median (interquartile range) (IQR), and n (%) for categorical variables. Differences in continuous variables were assessed using 2 sample t-test when normality and homogeneity assumptions were satisfied; otherwise, Mann-Whitney U test was used where data was not distributed normally. Chi-square or Fisher’s exact test was used for categorical variables. Covariates that were selected a priori for variable adjustment were gender, age, hypertension, atrial fibrillation, large vessel occlusion, diabetes mellitus, admitting NIHSS and admitting SBP. Logistic regression assessed the associations between lipid levels and outcomes, and with LAA. Results were presented as adjusted odds ratio (adjOR) with 95% confidence interval (CI). Statistical significance was set at two-sided *p* < 0.05.

## 3. Results

This section may be divided by subheadings. It should provide a concise and precise description of the experimental results, their interpretation, as well as the experimental conclusions that can be drawn.

### 3.1. Baseline Characteristics and Associations of Lipid Parameters with LAA

1265 consecutive patients with ischemic stroke treated with IV tPA were analyzed and 1004 patients with valid lipid assessments were included in this study. Of 1004 patients, 589/986 (59.7%) were male, 586/883 (66.4%) were of Chinese ethnicity, 596/916 (65.1%) experienced a LVO and 190/1004 (18.9%) had AF. There were 657/1004 (65.4%) patients with history of with hypertension, 526/1004 (52.4%) with hyperlipidemia, and 306/1004 (30.5%) with diabetes mellitus. Median age was 66 (IQR 56–77) years while median admitting NIHSS was 15 (IQR 8–21). Median LDL-C, non-HDL-C, TC, HDL-C and LDL-C/HDL-C were 2.86 (IQR 2.18–3.50) mmol/L, 3.43 (IQR 2.70–4.19) mmol/L, 4.62 (IQR 3.86–5.36) mmol/L, 1.12 (IQR 0.95–1.32) mmol/L and 2.49 (IQR 1.84–3.30) respectively. In accordance with the TOAST classification, 322/975 (33.0%) had LAA stroke, 341/975 (35.0%) had cardioembolic (CE) stroke, 168/975 (17.2%) had small vessel occlusion (SVO), 10/975 (1.0%) had stroke of other determined etiology and 134/975 (13.7%) had cryptogenic stroke (Table 1).

Comparing LAA and non-LAA groups, median LDL-C, non-HDL-C, LDL-C/HDL-C ratio, as well as white blood cell (WBC) count, neutrophils, platelets, admitting NIHSS and presence of LVO were significantly higher in the LAA compared to non-LAA group (Appendix A). Regarding lipid parameters and associations with LAA, after adjustment for gender, age, hypertension, atrial fibrillation, large vessel occlusion, diabetes mellitus, admitting NIHSS and admitting SBP, the following results were obtained. Q4 of LDL-C was associated with increased odds of LAA (adjusted odds ratio (adjOR) 1.69, 95% CI 1.07–2.69, *p* = 0.024) compared to Q1. Q2 and Q4 of non-HDL-C was significantly associated with increased odds of LAA (adjOR 1.64, 95% CI 1.04–2.60, *p* = 0.035 and adjOR 1.75, 95% CI 1.10–2.80, *p* = 0.018 respectively) compared to Q1. Q3 of HDL-C was associated with increased odds of LAA (adjOR 1.84, 95% CI 1.16–2.95, *p* = 0.010) compared to Q4. Lastly, Q3 and Q4 of LDL-C/HDL-C were associated with increased odds of LAA (adjOR 1.88, 95% CI 1.18–3.00, *p* = 0.008 and adjOR 1.71, 95% CI 1.07–2.77, *p* = 0.027 respectively) compared to Q1 (Table 2).

### 3.2. Stroke Outcomes

Of 1004 patients, 479/995 (48.1%) experienced poor functional outcomes (mRS 3–6), 48/1003 (4.8%) suffered SICH and 117/990 (11.8%) died. (Table 3) There was no statistically significant difference in prevalence of these three outcomes between the LAA and non-LAA group (Appendix A).

### 3.3. Associations of Lipid Parameters with Poor Functional Outcome, SICH and Mortality

When evaluating the following relationships between lipid parameters and outcomes measured, variables adjusted for in the multivariate model include gender, age, hypertension, atrial fibrillation, large vessel occlusion, diabetes mellitus, admitting NIHSS and admitting SBP. 

#### 3.3.1. Poor Functional Outcome

Restricted cubic spline curves showed a ‘reverse tick’ relationship between LDL-C, non-HDL-C and TC with poor functional outcome (Figure 2).

In multivariate analysis, Q1, Q2 and Q3 of HDL-C were significantly associated with increased odds of poor functional outcome (adjOR 1.66, 95% CI 1.06–2.60, *p* = 0.028, adjOR 1.63, 95% CI 1.05–2.53, *p* = 0.027 and OR 1.56, 95% CI 1.01–2.44, *p* = 0.048 respectively) when compared to Q4. Q2 and Q4 of LDL-C/HDL-C ratio were associated increased odds of poor functional outcome (adjOR 1.56, 95% CI 1.02–2.41, *p* = 0.043, OR: 1.78, 95% CI 1.16–2.76, *p* = 0.009 respectively) when compared to Q3 (Table 4).

#### 3.3.2. SICH

Restricted cubic spline curve showed a U-shaped association between non-HDL-C and LDL-C with SICH. (Figure 2) The lipid parameters that had significant non-linear relationships with SICH were non-HDL-C and TC. In multivariate analysis, Q2 and Q4 of non-HDL-C were significantly associated with SICH (adjOR 4.28, 95% CI 1.36–18.90, *p* = 0.025 and adjOR 5.17, 95% CI 1.64–22.81, *p* = 0.011 respectively) when compared to Q3. Similarly, Q2 and Q4 of TC were significantly associated with increased odds SICH (adjOR 4.46, 95% CI 1.43–19.59, *p* = 0.021 and adjOR 5.29, 95% CI 1.69–23.33, *p* = 0.010 respectively) when compared to Q3. Q2 of HDL-C remained significantly associated with increased odds of SICH (adjOR 3.07, 95% CI 1.23–8.75, *p* = 0.022) when compared to Q1 (Table 4).

#### 3.3.3. 90-Day Mortality

Restricted cubic spline curves showed a ‘reverse tick’ relationship of LDL-C, non-HDL-C and TC with mortality. (Figure 2) In multivariate analysis, Q1 and Q2 of LDL-C were significantly associated with increased odds of mortality (adjOR 2.57, 95% CI 1.27–5.57, *p* = 0.011 and adjOR 2.28, 95% CI 1.10–5.02, *p* = 0.032) when compared to Q3. Q2 of LDL-C/HDL-C was associated with increased odds of mortality (adjOR 2.51, 95% CI 1.30–5.05, *p* = 0.008) when compared to Q3. No significant associations were found to relate non-HDL-C, TC and HDL-C with mortality on multivariate analysis (Table 4). Results of logistic regressions that adjusted for age and gender only, and age, gender, admitting NIHSS and LVO, are illustrated in the Appendix A. 

## 4. Discussion

It was found that firstly, low HDL-C was associated with poor functional outcome, and secondly, a U-shaped relationship was found between non-HDL-C and SICH. Finally, low LDL-C was also found to be associated with increased mortality.

### 4.1. Lipid Parameters and Functional Neurological Outcome

We found that high HDL-C tended to be protective against poor functional outcome. This corroborates with a past Japanese study that found higher HDL-C and increased odds of favourable functional outcome after tPA thrombolysis [15]. This can be explained by how HDL reduces neuronal injury after ischemic stroke, possibly through anti-oxidative or anti-inflammatory pathways [16]. By suppressing inflammatory responses, HDL-C could promote neurological recovery because inflammatory cells and responses like neutrophils and neutrophil accumulation have been shown to cause poorer neurological outcome [17]. Next, our study found that the relationship between LDL-C/HDL-C and functional outcome may be non-linear, expanding on a previous Chinese study of 763 AIS patients treated with tPA, that demonstrated a LDL-C/HDL-C ratio cut-off of <2.71 was associated with higher risk of poor outcome [18]. Higher LDL-C/HDL-C was also found to be protective against mRS >2 at 3 months [19]. However, the relationship between LDL-C/HDL-C and functional outcome after ischemic stroke thrombolysis continues to be inconclusive.

### 4.2. Lipid Parameters and SICH

When investigating relationships with SICH, we found a U-shaped and non-linear relationship of non-HDL-C and TC with SICH respectively. Previously, a Chinese study found that lower non-HDL-C resulted in greater risk of hemorrhagic transformation [5], but other studies were unable to validate this association [3,20]. Conclusions on associations between TC and SICH were also varied [21,22]. Our finding of a U-shaped relationship between non-HDL-C and SICH where increased risk of SICH occurs at either moderately low or high non-HDL concentrations could explain the discrepancy in prior studies’ results. Postulated mechanisms include how low cholesterol reflects poor general health and undernourishment that predisposes individuals to hemorrhagic transformation unrelated to cholesterol pathways [23]. High non-HDL-C could increase risk of SICH through the development of arterial stiffness, which independently increases the risk of hemorrhagic transformation in thrombolysis-treated stroke patients [24]. Next, some studies have suggested a correlation between lower LDL-C and increased SICH risk [25,26]. In our study, non-HDL-C, but not LDL-C, was significantly associated with SICH. This is a surprising finding because LDL-C is a major component of non-HDL-C. However, other lipoproteins which contribute to non-HDL-C, such as very-low-density lipoproteins, intermediate-density lipoproteins and lipoprotein(a) could affect SICH occurrence. For instance, lipoprotein(a) was found to reduce bleeding risk in the brain due to its hemostatic properties [27]. Past studies have also demonstrated significant differences in the predictive ability of non-HDL-C and LDL-C on major adverse cardiac events (MACE) [28], reinstating the different roles of non-HDL-C and LDL-C in cardiovascular and cerebrovascular events.

### 4.3. LDL-C and Mortality

Our study found low LDL-C was significantly associated with increased odds of 90-day all-cause mortality after multivariate adjustment. This association has also been previously suggested outside of stroke—in a systematic review involving 68094 elderly patients, an inverse association between LDL-C and all-cause mortality was found, hypothesized by how low LDL-C increased vulnerability to fatal illnesses [29]. In our study, patients with low LDL-C had more comorbidities, (Appendix A) which likely increased mortality predisposition. Hence, it was proposed that low LDL-C is an indirect indicator of severe illness rather than the cause of increased mortality [30]. Another explanation, although speculative, is the interaction between low LDL-C with dysbiosis and changes to bile acid metabolism that eventually leads to mortality [31]. The finding of low LDL-C increasing mortality may also be attributed to AIS patients without prior statin use, supported by Cheng et al. who found that low LDL-C was associated with higher mortality rates in statin-naïve acute ischemic stroke [32]. This finding is therefore generalized without comparison between statin, non-statin or proprotein convertase subtilisin/kexin type 9 (PCSK9) inhibitor use.

### 4.4. Analysis of Lipid Parameters with LAA Subtype

This study also found that LDL-C, non-HDL-C, HDL-C and LDL-C/HDL-C were significantly associated with LAA after multivariate adjustment. Previous studies found elevated LDL-C a risk factor for atherothrombotic infarct, and LDL-C higher in LAA than other stroke subtypes [33,34]. Our finding of high LDL-C and increased odds of LAA can be explained by LDL-C’s association with intracranial and extracranial stenosis [35]. Next, our study found that higher HDL-C may be more desirable than moderate HDL-C levels in protection against LAA, reasoned by how HDL-C increases LDL-C reverse transport, delivers antioxidants to LDL-C and decreases susceptibility of LDL-C to oxidation in endothelium, slowing the process of atherosclerosis [36]. Lastly, our finding of high LDL-C/HDL-C ratio and association with increased odds of LAA can be explained by the atherogenicity of LDL-C and atheroprotection by HDL-C, supporting the previous finding of LDL-C/HDL-C ratio and its association with increased intima-media thickness, a measure of subclinical atherosclerosis [37]. Thus, high LDL-C/HDL-C ratio may be a useful indicator of atherosclerosis to help identify patients at higher risk of LAA stroke. 

## 5. Strengths and Limitations

Our study is a comprehensive report detailing associations of selected lipid parameters with post-thrombolysis outcomes, representing a relatively large cohort size (1004 patients analysed) compared to previous thrombolysis studies. However, this study was a single institution retrospective cohort study that solely evaluated intravenous thrombolysis patients. This may limit the generalisability of results to other cohorts, warranting more prospective studies on lipid parameters and ischemic stroke outcomes. We would like to acknowledge the possibility of Type I error in the multivariate analyses in which significant associations may no longer hold true after Bonferroni correction. Excessive correction of statistical level of significance may also increase the likelihood of Type II error as a trade-off to reduce Type I error, which may increase false negatives. The multivariate relationship between HDL-C and poor functional outcome is to be interpreted with caution as no significant relationship was found on univariate analysis, in which the possible reasons can be explained statistically by Lo et al. and Wang et al. [38,39]. Hence further studies in this area are required to confirm this association. Our study did not distinguish statin and non-statin users or evaluated use of other lipid-lowering agents, nor compare antithrombotic drug use which could theoretically affect SICH risk. Comorbidities like chronic kidney disease could be a confounder that should be explored in future studies. We would suggest future studies compare ischemic stroke outcomes between intravenous thrombolysis, intra-arterial thrombolysis and mechanical thrombectomy cohorts, and evaluate the role of other non-traditional lipid measures like TG/HDL-C and TC/HDL-C ratio in these treatment options.

## 6. Conclusions

In AIS patients who received IV tPA, low LDL-C was associated with increased odds of mortality while HDL-C may be protective against poor functional outcome.

## Figures and Tables

**Figure 1 jcm-11-07148-f001:**
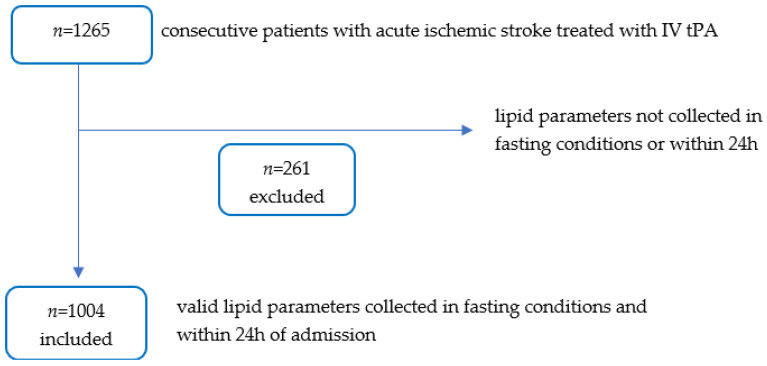
Flow Diagram Illustrating Only Patients with Valid Lipid Parameters were Analysed.

**Figure 2 jcm-11-07148-f002:**
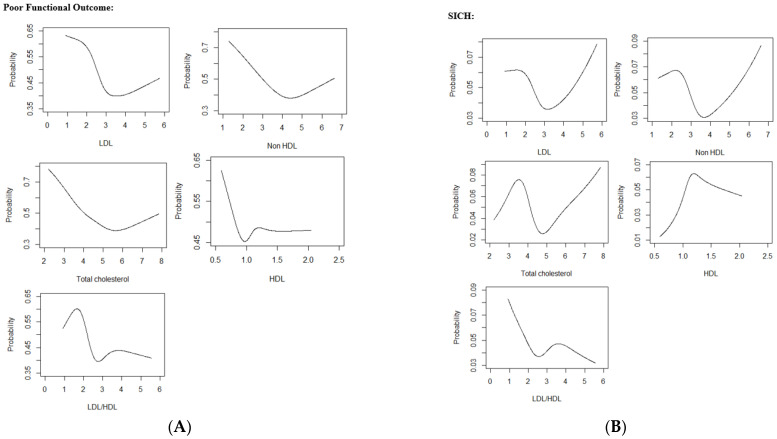
(**A**) Restricted Cubic Spline Curves relating Lipid Parameters with Poor Functional Outcome (**B**) Restricted Cubic Spline Curves relating Lipid Parameters with SICH (**C**) Restricted Cubic Spline Curves relating Lipid Parameters with Mortality.Abbreviations: LDL low-density lipoprotein cholesterol, HDL high-density lipoprotein cholesterol, LDL/HDL low-density lipoprotein cholesterol/high-density lipoprotein cholesterol ratio, SICH symptomatic intracranial hemorrhage.

**Table 1 jcm-11-07148-t001:** Baseline Characteristics of Study Population.

	Total (*n* = 1004)
Age (years)	66 [56,77]
Gender (male)	589/986 (59.7)
Race	
Chinese Malay Indian Others	586/883 (66.4) 188/883 (21.3) 62/883 (7.0) 47/883 (5.3)
Lipid parameters	
LDL-C (mmol/L)	2.86 [2.18, 3.50]
Non-HDL-C (mmol/L)	3.43 [2.70, 4.19]
HDL-C (mmol/L)	1.12 [0.95, 1.32]
TC (mmol/L)	4.62 [3.86, 5.36]
LDL-C/HDL-C Ratio	2.49 [1.84, 3.30]
Stroke parameters	
Admitting NIHSS	15 [8,21]
Admitting SBP (mmHg)	152 [136,168]
Admitting DBP (mmHg)	82 [72,92]
Large vessel occlusion	596/916 (65.1)
Comorbidities	
Smoker	148 (14.7)
Hypertension	657 (65.4)
Hyperlipidemia	526 (52.4)
Diabetes mellitus	306 (30.5)
Atrial fibrillation	190 (18.9)
Stroke Subtype (TOAST)	
LAA CE SVO Stroke of other determined etiology Cryptogenic	322/975 (33.0) 341/975 (35.0) 168/975 (17.2) 10/975 (1.0) 134/975 (13.7)
Glucose level	
Fasting glucose (mmol/L)	5.90 [5.20, 7.30]
HbA1C (%)	6.00 [5.60, 6.90]

Values are median [IQR] for numerical variables and n/total (%) or n (%) for categorical variables. Abbreviations: mRS modified Rankin Scale, LDL-C high-density lipoprotein cholesterol, non-HDL-C non-high-density lipoprotein cholesterol, TC total cholesterol, HDL-C high-density lipoprotein cholesterol, NIHSS National Institutes of Health Stroke Scale, SBP systolic blood pressure, DBP diastolic blood pressure, TOAST Trial of ORG 10172 in Acute Stroke Treatment, LAA large-artery atherosclerosis, CE cardioembolic, SVO stroke of other determined etiology, HbA1c Hemoglobin A1c.

**Table 2 jcm-11-07148-t002:** Lipid Parameters and Associations with LAA.

	LAA			
	Univariate Analysis		Multivariate Analysis	
	OR (95% CI)	*p*-Value	OR (95% CI)	*p*-Value
LDL-C: Q1 Q2 Q3 Q4	1.0 1.28 (0.87–1.88) 1.03 (0.69–1.51) 1.64 (1.13–2.40)	- 0.209 0.898 0.010	1.0 1.42 (0.90–2.25) 1.27 (0.79–2.05) 1.69 (1.07–2.69)	- 0.130 0.323 0.024
Non-HDL-C: Q1 Q2 Q3 Q4	1.0 1.39 (0.95–2.04) 1.09 (0.74–1.62) 1.62 (1.11–2.37)	- 0.093 0.657 0.013	1.0 1.64 (1.04–2.60) 1.25 (0.78–2.01) 1.75 (1.10–2.80)	- 0.035 0.355 0.018
TC: Q1 Q2 Q3 Q4	1.03 (0.70–1.52) 1.0 1.11 (0.76–1.62) 1.30 (0.89–1.90)	0.874 - 0.603 0.170	0.93 (0.59–1.47) 1.0 1.11 (0.71–1.74) 1.28 (0.82–1.99)	0.767 - 0.645 0.277
HDL-C Q1 Q2 Q3 Q4	1.44 (0.98–2.13) 1.30 (0.88–1.93) 1.76 (1.19–2.60) 1.0	0.067 0.183 0.005 -	1.27 (0.79–2.04) 1.30 (0.81–2.08) 1.84 (1.16–2.95) 1.0	0.320 0.276 0.010 -
LDL-C/HDL-C: Q1 Q2 Q3 Q4	1.0 1.10 (0.74–1.63) 1.67 (1.14–2.46) 1.69 (1.16–2.49)	- 0.646 0.009 0.007	1.0 1.26 (0.79–2.02) 1.88 (1.18–3.00) 1.71 (1.07–2.77)	- 0.333 0.008 0.027

Abbreviations: OR: Odds Ratio, 95% CI 95% Confidence Interval, Q1 first quartile, Q2 s quartile, Q3 third quartile, Q4 fourth quartile, LAA large-artery atherosclerosis, LDL-C high-density lipoprotein cholesterol, non-HDL-C non-high-density lipoprotein cholesterol, TC total cholesterol, HDL-C high-density lipoprotein cholesterol. Variables adjusted for in multivariate analysis include gender, age, hypertension, atrial fibrillation, large vessel occlusion, diabetes mellitus, admitting NIHSS and admitting SBP. The quartiles for non-HDL-C were Q1: ≤2.7mmol/L, Q2: >2.7–3.43 mmol/L, Q3: >3.43–4.19 mmol/L, Q4: >4.19mmol/L. The quartiles for LDL-C were Q1: ≤2.18 mmol/L; Q2: >2.18–2.86 mmol/L, Q3: >2.86–3.50 mmol/L, Q4: >3.50 mmol/L. The quartiles for TC were Q1: ≤3.85 mmol/L, Q2: >3.85–4.63 mmol/L, Q3: >4.63–5.36 mmol/L, Q4: >5.36 mmol/L. The quartiles for HDL-C were Q1: ≤0.95 mmol/L, Q2: >0.95–1.12 mmol/L, Q3: >1.12–1.32 mmol/L, Q4: >1.32 mmol/L. The quartiles for LDL/HDL ratio were Q1: ≤1.84, Q2: >1.84–2.49, Q3: >2.49–3.30, Q4: ≥3.30.

**Table 3 jcm-11-07148-t003:** Stroke Outcomes.

Stroke Outcomes	Total (*n* = 1004)
Poor functional outcome (90-day mRS 3–6)	479/995 (48.1)
SICH	48/1003 (4.8)
90-day mortality	117/990 (11.8)

Values are *n*/total (%). Abbreviations: mRS modified Rankin Scale, SICH symptomatic intracranial hemorrhage.

**Table 4 jcm-11-07148-t004:** Association of Lipid Parameters with Poor Functional Outcome, SICH and Mortality.

	Poor Functional Outcome (mRS 3–6)					
	Univariate Analysis			Multivariate Analysis		
	OR	95% CI	*p*-Value	OR	95% CI	*p*-Value
LDL-C: Q1 Q2 Q3 Q4	2.18 1.19 1.00 1.0	1.52–3.12 0.83–1.70 0.70–1.43 -	<0.001 0.339 0.997 -	1.33 1.0 1.11 1.12	0.87–2.03 - 0.72–1.70 0.73–1.71	0.196 - 0.647 0.607
Non-HDL-C: Q1 Q2 Q3 Q4	2.56 1.51 1.0 1.16	1.79–3.68 1.06–2.15 - 0.81–1.66	<0.001 0.024 - 0.421	1.30 1.05 1.0 1.19	0.84–2.03 0.69–1.62 - 0.78–1.83	0.241 0.809 - 0.416
TC: Q1 Q2 Q3 Q4	2.38 1.37 1.0 1.08	1.66–3.42 0.96–1.95 - 0.76–1.54	<0.001 0.082 - 0.672	1.38 1.05 1.0 1.10	0.90–2.14 0.69–1.62 - 0.72–1.68	0.141 0.810 - 0.651
HDL-C: Q1 Q2 Q3 Q4	1.09 1.09 1.0 1.04	0.77–1.56 0.76–1.54 - 0.72–1.49	0.626 0.646 - 0.839	1.66 1.63 1.56 1.0	1.06–2.60 1.05–2.53 1.01–2.44 -	0.028 0.027 0.048 -
LDL-C/HDL-C Ratio: Q1 Q2 Q3 Q4	2.31 2.00 1.0 1.39	1.61–3.32 1.40–2.87 - 0.98–2.00	<0.001 <0.001 - 0.069	1.27 1.56 1.0 1.78	0.81–1.97 1.02–2.41 - 1.16–2.76	0.294 0.043 - 0.009
	Symptomatic Intracranial Hemorrhage (SICH)					
	Univariate Analysis			Multivariate Analysis		
	OR	95% CI	*p*-value	OR	95% CI	*p*-value
LDL-C: Q1 Q2 Q3 Q4	2.37 1.62 1.0 2.10	0.99–6.27 0.63–4.48 - 0.86–5.63	0.062 0.324 - 0.115	1.84 1.41 1.0 2.22	0.71–5.37 0.52–4.21 - 0.85–6.47	0.227 0.514 - 0.117
Non-HDL-C: Q1 Q2 Q3 Q4	3.59 3.88 1.0 3.93	1.27–12.80 1.38–13.75 - 1.40–13.93	0.026 0.017 - 0.016	3.19 4.28 1.0 5.17	0.99–14.30 1.36–18.90 - 1.64–22.81	0.078 0.025 - 0.011
TC: Q1 Q2 Q3 Q4	4.92 5.28 1.0 5.68	1.58–21.55 1.71–22.98 - 1.86–24.62	0.013 0.009 - 0.006	3.58 4.46 1.0 5.29	1.12–15.96 1.43–19.59 - 1.69–23.33	0.052 0.021 - 0.010
HDL-C: Q1 Q2 Q3 Q4	1.0 2.82 1.82 2.68	- 1.15–7.93 0.67–5.43 1.07–7.63	- 0.032 0.253 0.045	1.0 3.07 2.08 2.26	- 1.23–8.75 0.74–6.34 0.85–6.74	- 0.022 0.173 0.116
LDL-C/HDL-C Ratio: Q1 Q2 Q3 Q4	2.23 1.40 1.0 1.54	0.97–5.54 0.56–3.67 - 0.62–3.99	0.068 0.479 - 0.355	1.49 1.07 1.0 1.55	0.63–3.83 0.41–2.89 - 0.61–4.14	0.378 0.897 - 0.367
	Mortality					
	Univariate Analysis			Multivariate Analysis		
	OR	95% CI	*p*-value	OR	95% CI	*p*-value
LDL-C: Q1 Q2 Q3 Q4	3.43 2.19 1.0 1.46	1.93–6.42 1.18–4.19 - 0.75–2.89	<0.001 0.015 - 0.272	2.57 2.28 1.0 1.96	1.27–5.57 1.10–5.02 - 0.89–4.49	0.011 0.032 - 0.101
Non-HDL-C: Q1 Q2 Q3 Q4	2.25 1.94 1.0 1.01	1.29–4.03 1.10–3.51 - 0.53–1.94	0.005 0.025 - 0.968	1.06 1.55 1.0 1.10	0.54–2.14 0.79–3.14 - 0.53–2.34	0.669 0.204 - 0.744
TC: Q1 Q2 Q3 Q4	2.60 1.65 1.0 1.08	1.50–4.63 0.92–3.02 - 0.57–2.05	<0.001 0.097 - 0.818	1.44 1.16 1.0 1.04	0.77–2.76 0.60–2.30 - 0.51–2.15	0.266 0.657 - 0.904
HDL-C: Q1 Q2 Q3 Q4	1.19 1.36 1.0 1.16	0.67–2.11 0.78–2.38 - 0.65–2.07	0.553 0.280 - 0.623	1.16 1.38 1.10 1.0	0.62–2.21 0.75–2.55 0.57–2.10 -	0.639 0.296 0.784 -
LDL-C/HDL-C Ratio: Q1 Q2 Q3 Q4	2.28 2.27 1.0 1.06	1.29–4.15 1.29–4.13 - 0.55–2.06	0.005 0.006 - 0.857	1.55 2.51 1.0 1.46	0.80–3.14 1.30–5.05 - 0.68–3.17	0.204 0.008 - 0.338

Abbreviations: OR odds ratio, 95% CI 95% confidence interval, SICH symptomatic intracranial hemorrhage, mRS modified Rankin Scale, Q1: first quartile, Q2: second quartile, Q3: third quartile and Q4: fourth quartile. LDL-C high-density lipoprotein cholesterol, non-HDL-C non-high-density lipoprotein cholesterol, TC total cholesterol, HDL-C high-density lipoprotein cholesterol. Variables adjusted for in multivariate analysis include gender, hypertension, atrial fibrillation, large vessel occlusion, diabetes mellitus, age, admitting NIHSS and admitting SBP. The quartiles for non-HDL-C were Q1: ≤2.7mmol/L, Q2: >2.7–3.43 mmol/L, Q3: >3.43–4.19mmol/L, Q4: >4.19mmol/L. The quartiles for LDL-C were Q1: ≤2.18mmol/L; Q2: >2.18–2.86 mmol/L, Q3: >2.86–3.50 mmol/L, Q4: >3.50 mmol/L. The quartiles for TC were Q1: ≤3.85mmol/L, Q2: >3.85–4.63mmol/L, Q3: >4.63–5.36mmol/L, Q4: >5.36mmol/L. The quartiles for HDL-C were Q1: ≤0.95 mmol/L, Q2: >0.95–1.12 mmol/L, Q3: >1.12–1.32 mmol/L, Q4: >1.32 mmol/L. The quartiles for LDL/HDL ratio were Q1: ≤1.84, Q2: >1.84–2.49, Q3: >2.49–3.30, Q4: ≥3.30.

## Data Availability

Not applicable.

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
