# Peer review of "Impact of Traditional and Non-Traditional Lipid Parameters on Outcomes after Intravenous Thrombolysis in Acute Ischemic Stroke"

_jcm, 2022, doi:10.3390/jcm11237148_

Round 1
Reviewer 1 Report
I would suggest changing the title of the paper (Impact of Traditional and Non-traditional Lipid Parameters...).
In Discussion, the authors should review their findings that low LDL-C was associated with increased odds of mortality in context of the results of large clinical trials of lipid-lowering agents and provide a plausible explanation for their results. Otherwise, it could be misleading.
In addition, it would be interesting to know what proportion of the sample had been taking statins before the stroke, if this information is available.
Reviewer 2 Report
I reviewed an article entitled “Traditional and Non-traditional Lipid Parameters on Outcomes 2 after Intravenous Thrombolysis in Acute Ischemic Stroke”. The manuscript was well written. I have some comments.
1. Please provide flow diagram of participants selection based on inclusion and exclusion criteria.
2. Table 1 should be revised, title of characteristics (e.g. race, lipid parameters…..) should be written on the left-hand side.
3. Median age (line 118) should be written with unit (years?).
4. Why the number of patients in Table S1 summed up to 975 not 1004?
5. Authors should analyze the data using more than 1 multivariable regression model. I’m not sure that the association found in this manuscript was the true association or not.
6. Authors should consider chronic kidney disease is one of the confounders.
Round 2
Reviewer 2 Report
I am satisfied with authors' responses.